# Interceptive Treatment with Invisalign^®^ First in Moderate and Severe Cases: A Case Series

**DOI:** 10.3390/children9081176

**Published:** 2022-08-05

**Authors:** Teresa Pinho, Duarte Rocha, Sofia Ribeiro, Francisca Monteiro, Selma Pascoal, Rui Azevedo

**Affiliations:** 1UNIPRO—Oral Pathology and Rehabilitation Research Unit, University Institute of Health Sciences (IUCS), Cooperativa de Ensino Superior Politécnico e Universitário (CESPU), 4585-116 Gandra, Portugal; 2IBMC—Instituto Biologia Molecular e Celular, i3S—Instituto de Inovação e Investigação em Saúde, Universidade do Porto, 4200-135 Porto, Portugal; 3Center for MicroElectroMechanical Systems (CMEMS), University of Minho, 4800-058 Guimarães, Portugal; 4ICVS/3B’s—PT Government Associate Laboratory, University of Minho, 4710-057 Braga, Portugal; 5LABBELS—Associate Laboratory, University of Minho, 4800-058 Guimarães, Portugal; 6TOXRUN-Toxicology Research Unit, University Institute of Health Sciences, Cooperativa de Ensino Superior Politécnico e Universitário (CESPU), 4585-116 Gandra, Portugal

**Keywords:** clear aligners, early diagnosis and treatment, interceptive orthodontic treatment, Invisalign^®^ First

## Abstract

The increasing demand for more aesthetic/comfortable orthodontic alternatives fostered the utilization of clear aligners in recent years. However, the efficacy of clear aligners for treating complex malocclusions is often treated with scepticism. This case series aims to evaluate the predictability of the Invisalign^®^ First system in moderate and severe cases requiring interceptive orthodontic treatments in mixed dentition. A total of 23 patients with 102 interceptive orthodontic malocclusion traits were selected for orthodontic treatment with Invisalign^®^ First and were examined over 18 months (Phase 1). Clinical assessments included ClinCheck^®^ predictions, cephalometric measurements, and measuring tools commonly used to quantify tooth movement. Measurements taken at the beginning and end of the treatment were compared. The complexity degree of each case was established based on the set of problems presented by each patient. All treatment objectives were achieved within 18 months, except for two Class II cases, with 69% of them solved with the first set of aligners. Additional aligners were used in the remaining cases. Even though these 23 cases suggest that the Invisalign^®^ First (Phase 1) may be effective in most interceptive problems, controlled randomized clinical trials are required to evaluate movement predictability and how this relates to the problem complexity and additional aligners required.

## 1. Introduction

Interceptive orthodontic problems include functional oral complications, pathologies, and/or malocclusion traits that usually occur in mixed dentition, at an early age and before growth is complete, and that can evolve into serious and complicated orofacial malocclusion traits in adulthood if not treated in time. Early diagnosis is essential for preventing future extensive orthodontic treatment. Interceptive orthodontic interventions in early mixed dentitions have a common objective of allowing a more favorable functional growth of the jaws and the consequent development of a satisfactory occlusion [1]. For such treatments, the most common malocclusion traits to intercept are arch constriction, molar rotation, and crowding, which are commonly associated with a posterior crossbite, impacted teeth, and a loss of space for permanent dentition. Additionally, molar sagittal Class II or III and open bite are important to intercept to avoid the progression of skeletal malocclusion traits that may require future surgical intervention [2,3]. Furthermore, the early treatment of midline deviation prevents the development of skeletal and/or dental asymmetries [4,5,6].

Invisalign^®^ First systems, implemented in 2018 by Align Technology^®^, allow an interceptive treatment of 18 months (Phase 1) and the inclusion of an unlimited number of additional aligners planned by the ClinCheck^®^ software (ClinCheck Pro version 1.10 Align Technolgy Inc. San José, CA, USA). This involves using digital set-ups that allow the pediatric orthodontist to virtually plan each tooth’s movement in 3D [7,8]. Importantly, after ending Phase 1, the orthodontic treatment can be re-activated with a new treatment (Phase 2) with a duration of three years maximum. This could occur within ten years since Phase 1 ends if the malocclusion trait was not resolved or in case of a recurrence. In the present study, the movement predictability was only assessed for Invisalign First during Phase 1.

As with any other technology, the number of Invisalign^®^ aligners prescribed depends on several factors, such as dental malposition type, location, and complexity [7], and sometimes the way the case evolves is unpredictable by itself independent of the approach.

Currently, there is sparse scientific knowledge and research about the predictability and effectiveness of Invisalign^®^ systems as corrective treatments in later mixed or adult dentition [9,10,11,12]. To the best of our knowledge, there is only one study evaluating the efficacy of dentoalveolar expansion movement with Invisalign^®^ First technology in mixed dentition [13]. Hence, there are no proven data about the predictability and effectiveness of interceptive treatments requiring other types of problems/movements. 

Therefore, this case series intends to compare and retrospectively evaluate the predictability and effectiveness of the Invisalign^®^ First system in orthodontic treatments in children with early mixed dentition. In addition, the system effectiveness was analyzed as a function of the malocclusion traits and complexity degree. The guardians and children filled out short questionnaires to ascertain if the treatment met their expectations in different domains.

## 2. Materials and Methods

### 2.1. Case Series

This study comprised a sample of 23 Caucasian children (13 female and 10 male) having 102 interceptive malocclusion traits. The children were recruited in two independent clinics of orthodontics—Clínica Médico Dentária de São João da Madeira and Clínica de Medicina Dentária Dr. Manuel Neves—from October 2018 to October 2019. All patients were treated with Invisalign^®^ First for 18 months (Phase 1) by an Invisalign^®^ Diamond provider and double specialist in Orthodontics and Odontopediatrics (TP).

Patients were selected according to the following inclusion criteria: individuals with early mixed dentition treated with Invisalign^®^ First who underwent treatment in both arches; children who completed an initial and final intraoral digital scan; and cases requiring at least one movement with intermediate to high complexity based on an adaptation of the Align^®^ protocol, the Invisalign^®^ evaluation tool. Children with previous/concomitant orthodontic treatments and having craniofacial malformations (including cleft lip or palate), history of dental trauma, oral neoformations, or other oral cavity pathologies were excluded.

### 2.2. Ethical Considerations

The study was accepted by the Ethics Committee of the University Institute of Health Sciences (reference 11/CE-IUCS/2020).

### 2.3. Orthodontic Intervention

The children were instructed to use each aligner for as many hours as possible (20–22 h/day) and to only remove them to eat and during oral hygiene practices. Each aligner is designed to produce a 0.25-mm translation and 1° of rotation movement for standard forces (Align 2022) [14]. The aligners were changed every seven days at the beginning of the treatments, as recommended by Align^®^ protocols. In more complex cases with more sequenced movements and more than 50 aligners on the first set, the change was performed twice a week (only if the patient revealed compliance and if the aligners were adjusted). The control consultations were carried out every four weeks (shorter intervals compared with the same treatment in adults). 

Elastics were used as auxiliaries from the beginning of the treatment in moderate and complex cases of posterior crossbite, as well as in complex Class II or Class III relations. However, in Class II division 2 situations, these were only used when an overjet was created.

Additional aligners were only required at the final stage of this study period (just closing of Phase 1) to conclude the active phase in specific cases (e.g., when the objectives were not achieved yet, to perform overcorrections, or to work as a retention method). Additional aligners were only used during sleep. 

### 2.4. Clinical Assessment

The interceptive orthodontic malocclusion traits described here were classified as predictable, intermediate, or difficult corrections based on the Align^®^ recommendations adapted for children’s growth (see the classification criteria described in Table 1). 

The tool made available by Invisalign^®^ to classify the cases (i.e., the Invisalign^®^ evaluation tool) resulted in the same classification obtained in our study. Please note that according to the Invisalign^®^ criteria, a patient only needs one complex problem to be classified as complex/severe. Because we are dealing with children in mixed dentition, with temporary teeth and whose craniofacial structure will grow and change over time, we believe this is not the best methodology for the classification of our cases of global complexity. Given this, we have considered a combination of factors that together can represent a more realistic classification. 

This classification was performed based on an adaptation of the Align^®^ protocol [14] to consider the potential growth of the children over time. The alterations to the Align classification protocol are described after each malocclusion trait. The following criteria were defined for each interceptive problem:Molar derotation—MDR: This was analyzed using the movement values planned in the initial and final ClinCheck^®^ data. Only upper first molars were considered in order to uniformize the data. Predictable movements required 15° to 30° movements; intermediate malocclusions needed a derotation higher than 30° till 40°; and malocclusion traits demanding more than 40° derotation were classified as difficult rectifications. Adaptation to the Align^®^ protocol: An additional margin of 10° was considered for each range since this movement is considerably easier in mixed dentitions since the second permanent molar has not erupted yet.Dentoalveolar expansion—DAE: This problem was evaluated through the real values of the maxillary first molar transversal distances on the initial and final arch width table on the ClinCheck^®^ treatment plan. Here, simple tooth expansion (3–4 mm) and negative molar torque is a predictable movement. An expansion movement of > 4 to 6 mm with negative molar torque is considered to be an intermediate correction. Alternatively, difficult movements were considered to be expansions greater than 6 mm or greater than 4 mm, with the treatment having a skeletal component with positive molar torque (normally is associated with transversal skeletal problems). Adaptation to the Align^®^ protocol: Tooth torque was considered since the children are in a growing stage and the palatal suture is not closed yet.Space recovery—SR: This classification resulted from mesiodistal width measurements and predictions of the space available for missing teeth using a millimetric scale and the initial and final ClinCheck^®^ scans. Adaptation to the Align^®^ protocol: This malocclusion trait is not present in the Invisalign^®^ evaluation tool, but it is important to consider particularly in this stage since permanent teeth are erupting.Molar sagittal malocclusion—MSC II or III: Because cephalometry is bidimensional, having left and right structures superimposed, ClinCheck^®^ and clinical intraoral photographs images were prioritized. Clinically, according to the Angle classification, we have defined incomplete Class II (half premolar) and complete Class II associated with functional deviation or molar ectopic eruption as predictable movements. Class III tendency was classified as intermediate, whereas complete Class II (one premolar discrepancy) or Class III were considered to be difficult movements (confirmed by ANB cephalometric measures). Adaptation to the Align^®^ protocol: Align^®^ uses the mesialization/distalization quantification. Here, as we are dealing with growing patients in mixed dentition, it is more pertinent to evaluate the molar sagittal relation.Posterior crossbite—PCB: PCB classification was done based on ClinCheck^®^ treatment plan images and intraoral photographs. Movements were classified as predictable when we have crossbite only on deciduous teeth. Intermediate movements were considered to be PCB presenting a dental component (i.e., negative torque) on the first permanent molar (as an end tooth position). Difficult movements were associated with PCB accompanied by positive torque on first permanent molar (as an end tooth position) and skeletal compression. Adaptation to the Align^®^ protocol: Align^®^ evaluates the complexity of the cases according to the number of teeth involved (if one, it is mild, if two, it is moderate, and if three or more, it is severe). Here, in addition to permanent teeth, patients also presented deciduous teeth with the first molar as an end tooth; therefore, these cases cannot be compared with (or classified as) adult dentition. As such, we have classified the cases differently if the tooth in crossbite is permanent or deciduous. We have also considered the necessity of torque movement, as described for dentoalveolar expansion.Open bite—OP: OP was assessed based on the initial and final cephalometry, the ClinCheck^®^ images, and the presence of dental or skeletal components. The planned amount of posterior intrusion and/or anterior extrusion was also considered, as recommended by the Align^®^ protocol. Due to their skeletal components (confirmed by FMA cephalometric measures), all open bite pathologies were associated with difficult malocclusion traits/movements. Adaptation to the Align^®^ protocol: This malocclusion trait was assessed as recommended by Align^®^, although we considered the presence of skeletal problems.Midline discrepancy—MD: MD was assessed through the millimetric scale obtained from the ClinCheck^®^ treatment plan. The movement was considered predictable when 1 to 2 mm movement was required. Movements were classified as having intermediate complexity when more than 2 mm and less than 3 mm was prescribed, and teeth needing more than 3 mm movement were considered to be difficult malocclusions. Adaptation to the Align^®^ protocol: MD is not considered by Align^®^. However, the existence of a MD requires an interceptive treatment in order to prevent serious future asymmetries and to understand the amount of space to recover due to the early loss of deciduous teeth.Crowding—CR: This malocclusion was evaluated using the negative dentomaxillary discrepancy (DDM) for early mixed dentition: When the required movement ranged from 3 to 6 mm, the problem was considered to be predictable. When the required translation exceeded 6 mm to 8 mm, it was classified as an intermediate correction. If more than 8 mm of translation was required, the movement was considered to be a difficult rectification. In addition, the need for rotation was considered: Rotations greater than 30 through 40° on lateral incisors or greater than 40 through 50° on central incisors were considered to be intermediate CR, while rotations greater than 40° on lateral incisors or greater than 50° on central incisors were considered to be difficult corrections. Adaptation to the Align^®^ protocol: In the early mixed dentition, in addition to the ectopic eruption of the first molars, crowding in the anterior region is highly frequent. Due to this fact, we considered anterior permanent teeth rotation since this is one of the main problems to solve associated with crowding. Thus, it is also important to consider the required tooth rotation since the need for great rotation movements increases the complexity of crowding.

Then, based on this qualitative classification, a score of 1, 2 or 3 was respectively assigned for predictable, intermediate or difficult movements, from which a quantitative global complexity classification was obtained for each case (i.e., for each child). The sum of the scores categorizes the cases into mild (scoring lower than 3), moderate (between 3 and 8), or severe (greater than 8) (see the global complexity scale calculation in Table 2, in the Results). For high scores on severe cases, fewer teeth were selected to be moved in each stage, with their movements more sequenced, in order to increase the predictability of the overall movements.

The predictability of the Invisalign^®^ First was assessed by comparing tooth position/movement at the beginning and at the end of the treatment. Two types of data were compared: (i) the planned movement table provided by ClinCheck^®^ software (ClinCheck Pro version 1.10 Align Technolgy Inc. San José, USA) and the real amount of movement experimentally measured.

In addition, multiple other methods were used to analyze the included cases and to quantify and classify the movements/corrections to be accomplished. This includes the virtual three-dimensional (3D) planning using the ClinCheck^®^ treatment plan software, intraoral photographs, and digital models obtained with the iTero^®^ intraoral scanner from each patient. ClinCheck^®^ measurement tools (i.e., the planned movement table, real measures with the millimetric scale, and arch width) and cephalometric measurements were also conducted. Although not all cases presented skeletal malocclusion traits, the evaluation of cephalometric parameters was conducted in all patients to verify the overall improvement of the malocclusions. ANB (skeletal convexity) and FMA (mandibular plane angle) were used to analyze the sagittal and vertical skeletal components, respectively, while overjet and overbite were selected for the sagittal and vertical dental components, respectively. Signs and symptoms of temporomandibular disorders were also clinically assessed.

Questionnaires with five response items were given to the children and their caregivers to demonstrate patients’ cooperation as well as to ascertain if there was an agreement between their responses in order to determine potential treatment compliance flaws. 

### 2.5. Statistical Analysis

The statistical analysis was carried out in IBM SPSS^®^ (IBM SPSS Statistics 28, IBM Corp., Armonk, NY, USA) and R software (R version 4.1.2, R Foundation for Statistical Computing, Vienna, Austria), and Microsoft Excel^®^ (Microsoft Excel for Microsoft 365 Version 2206, Microsoft Corporation, Redmond, WA, USA) was used for the data organization and graphical reporting. A descriptive analysis of the movement and cephalometry metrics of all children was performed. When comparing achieved goals as a function of the complexity degree per pathology, the one-sided Cochran–Armitage Trend test was used considering a Bonferroni correction on the significance level for multiple testing (standard statistical method for reducing the inflation of type 1 errors, i.e., false positives, with multiple comparisons). The Wilcoxon signed-rank test was used to compare cephalometry metrics (i.e., overjet, overbite, ANB, and FMA), derotation, expansion, and space recovery before and after treatment. All cephalometric measurements were validated by calculating the standard intra-investigator error deviation using the Dahlberg method [15]. The significance level was set at 0.05.

## 3. Results

### Case Series

A total of 23 Caucasian children (13 female and 10 male) having 102 interceptive orthodontic malocclusion traits in mixed dentition were included in this case report. The age of the patients ranged from 6.8 to 10.3 years old, with an average age of 8.3 ± 1.0 years old. The desired movements/corrections included molar derotation, dentoalveolar expansion, space recovery, midline discrepancy, molar sagittal malocclusion, posterior crossbite, open bite, and crowding. No signs or symptoms of temporomandibular dysfunctions were observed. All patients complied with the Invisalign^®^ treatment, and no dropouts were registered. 

All patients were treated with Invisalign^®^ First for 18 months (Phase 1). This phase attempts to solve malocclusion traits that would affect function, future occlusion with permanent dentition, and possibly skeletal development as well. Therefore, each patient can have one or more treatment goals with different levels of complexity and, consequently, distinct predictability. 

The global classification for the treatment complexity for each patient was mild, moderate, or severe based on the discrete interceptive malocclusion traits that the children presented (see Table 2). All the included cases were classified as having moderate (*N* = 5) or severe (*N* = 18) complexity. 

A total of 102 interceptive orthodontic malocclusion traits were found in this sample (see the comprehensive quantitative and qualitative evaluation of the defined objectives in Table 3). Each case is illustrated and briefly characterized in Appendix A, which presents the intraoral photographs taken before and after the interceptive treatment as well as the virtual ClinCheck^®^ predictions for the desired results to be achieved at the end of the treatment. 

Here, a statistical study of different dental components is presented (see the descriptive analysis of MDR, DAE, SR, and cephalometric data in Table 4, and Appendix A with all measurements). Specifically, the molar derotation metrics of the upper permanent first molars (teeth 16 and 26) from nine patients show that the planned movement at the last ClinCheck^®^ was greatly reduced compared with the initial ClinCheck^®^ prediction for both teeth. Additionally, the dentoalveolar expansion initially planned for teeth 16 to 26 in 20 patients was achieved at the end of the orthodontic intervention, as the mean, minimum, and maximum values demonstrate. Similarly, data from 16 patients requiring space opening for teeth 12, 15, 22, 32, 33, and 43 reveal that a great part of the amount of movement initially planned for these teeth was accomplished (more than 98%). Moreover, cephalometric metrics on the entire sample show that overbite and overjet malocclusion traits observed in 23 patients were mitigated by the end of the orthodontic treatment.

Overall, this preliminary analysis suggests that the interceptive treatment tended to normalize the dental component of the examined cases (as observed through overbite and overjet data), while ANB values tended to decrease, reflecting the worsening of the skeletal component in Class III situations and improvement in Class II cases. On the other hand, in general, FMA scores did not reveal a tendency to modify the skeletal component in the vertical direction. The intra-operator error associated with the cephalometric measurements was assessed using the Dahlberg method [15]. As shown in Appendix A the intra-operator error was very low for all cephalometric features and time points, reflecting the repeatability of the measurements. 

After this preliminary descriptive analysis, the most adequate statistical tests were conducted. The Wilcoxon signed-rank test was used to evaluate the progression in the cephalometry parameters, as well as the derotation, expansion, and space recovery metrics, from the beginning to the end of the study (see Table 5). If there was no change in the need for correction between the beginning and the end, it means that there were no improvements (reflected by similar numbers in negative/positive ranks). In Table 5, it is demonstrated that most of the values related to the need for correction decreased significantly (reflected by the predominance of negative ranks).

Statistically significant differences were found for all metrics, except for FMA. This means that the tendency for a decrease in ANB data is statistically significant. Furthermore, final and initial FMA scores were not significantly different, which is in line with the data from Table 4. 

Then, we decided to ascertain if the total number of aligners required by the patients varied depending on the malocclusion trait complexity and the proposed treatment plan. Despite the presence of eruption compensations, which is a valuable tool for these treatments with Invisalign^®^ in mixed dentition, it is not possible to fully control the permanent teeth eruption, which increased the probability of misaligned aligners. Due to this factor, seven patients had several aligners planned that were never used. After outlier removal, the average difference between the planned and used aligners for these patients was approximately seven.

The most frequently diagnosed situation was arch constriction and, consequently, the need for dentoalveolar expansion. The goals defined for the interceptive treatment were achieved, and the malocclusion traits were solved within 18 months for all patients, except for two cases of MSC malocclusion, which was successful in only 89% of the cases. However, only 69% of the interceptive malocclusion traits could be solved with the initial series of aligners (see Figure 1). 

While more than 80% of the malocclusion traits related to MDR, DAE, SR, and CR were solved with no refinements, fewer than 55% of the cases with MSC, PC, OB, and MD did not need additional aligners. Particularly, none of the open bite cases was corrected with the first set of aligners.

Moreover, the only statistically significant relationship between the initially established degree of complexity and the percentage of the interceptive malocclusion traits being solved was found for MSC malocclusions at the end of the first set of aligners (*p*-value = 0.0035), with only 14% (1 out of 7) of the objectives achieved for the MSC malocclusion traits classified as difficult movements (see Table 6 and Figure 1).

The relationship between the global complexity of the cases and the number of additional aligners required was also analyzed (see Figure 2). All cases were classified as moderate (*N* = 5) or severe (*N* = 18). Four moderate and four severe cases did not need additional aligners. However, one moderate case required two refinements, while all the other cases needing additional sets of aligners had been previously classified as severe. As previously stated, two MSC cases did not complete the interceptive treatment within the study span (18 months; Invisalign^®^ First Phase 1). Although it was not statistically significant, a higher number of additional aligners was necessary in severe cases. 

Moreover, Figure 3 confirms that the malocclusion traits showing a high percentage of accomplishment after the first set of additional aligners (i.e., MDR, DAE, and SR) were the less severe traits. More specifically, severe cases usually coincide with the need for more than one additional aligner series, while all moderate cases required no refinements (Figure 2), with the exception of case 14, who presented the need for DAE (predictable problem), space recovery (intermediate complexity), and solving a MSC malocclusion (difficult rectification). For instance, the three open bite cases described here were initially classified as difficult movements, and none of them was concluded with the first set of aligners. Additionally, the only two interceptive malocclusion traits that were not concluded within the study span—two MSC malocclusions—presented a high prevalence of intermediate (33%) and difficult malocclusion traits (39%) compared with most of the other malocclusion traits (Figure 3). As expected, these data show that there is a relationship between the number of refinements required and the complexity of each case. 

Finally, according to the results of the questionnaires administered to the patients and their caregivers, all subjects reported that the aligners were used for the period that the orthodontist determined, and the vast majority only removed them during meals (Table 7). Some of the surveyed children removed the aligners in situations not indicated by the orthodontist, such as at school (*N* = 2), a situation in which there is no guardian control, or sometimes during sleep (*N* = 1). None of these sporadic situations were associated with the two unsolved MSC pathologies. There was a difference between the children and their guardians regarding whether the children liked using the aligners, although it was not statistically significant. To the question “Does the child like to use aligners?”, 87% (20 out of 23) of the guardians answered “Yes” while only 70% (16 out of 23) of the children agreed with this statement (Table 7). This discrepancy did not have an impact on treatment adherence.

## 4. Discussion

In the present study, 69% of the malocclusion traits (70 out of 102 interceptive problems) were solved with the first set of aligners, with aligner change every week, considering Align Technology^®^’s recommendations. However, for some difficult malocclusion traits requiring highly sequenced movements (and, consequently, a greater number of aligners compared with simpler cases), the aligner was changed twice a week only when the child was perfectly adapted to the orthodontic stimulus. This proved to be useful for achieving the desired results more quickly, especially when a large number of aligners was required. 

Ideally, additional aligners would not be necessary, and all objectives should be achieved in the initial aligners stage [16,17]. However, in clinical practice, this is not always possible. For some cases, the entire sequence of aligners was not used, requiring additional aligners for readjustment due to deciduous exfoliation, permanent teeth eruption, and some maladjustments when the real movement was not according to that planned by ClinCheck^®^ software (ClinCheck Pro version 1.10 Align Technolgy Inc. San José, CA, USA) [17]. Importantly, intraoral scans were the main priority to analyze here, and the intraoral photographs were evaluated to confirm the bite position of the digital models in occlusion.

Previous studies in later mixed and adult dentition shows that one of the least accurate tooth movements with Invisalign^®^ is incisor extrusion [7,18]. In this study, the cases that required this type of movement were associated with open bite malocclusions, requiring more aligners than other malocclusion traits due to movement unpredictability. However, the treatment of open bite malocclusion was still successful within the 18 months period.

Skeletal open bite cases are best addressed with posterior intrusionREF [19]. However, in our three cases, anterior extrusion was also needed, due to an inverted and lower smile. Although we had only three cases of open bite, these were within a complex severity with a skeletal component of significant hyperdivergence but were successfully intercepted in this first treatment phase with aligners. In the predictable open bite, interceptive treatment with myofunctional therapy may be further applied to treat and stabilize the teeth, even after finishing the active orthodontic treatment, for example making use of Trainer for Kids^®^ therapy [20].

Cephalometric measurements of the skeletal component in the sagittal direction, namely ANB, revealed that the orthodontic intervention did not enhance Class III malocclusions, although it improved Class II cases. This might be justified due to the plastic interposed between the arches that allows for posterior intrusion and consequently leads to the anterior rotation of the mandible, beneficial in hyperdivergent cases but not in hypodivergent biotypes [21]. In the same study, the FMA data revealed that hypodivergent cases worsened, while hyperdivergent biotypes were reduced by the end of the intervention, although the use of mini-screws could have helped posterior intrusion [21]. Contrarily, in the present case series on interceptive treatment in mixed dentition, no significant differences were found between FMA scores at the initial and final assessments. Nevertheless, our data show that Invisalign^®^ First did not aggravate the skeletal component in the vertical direction. 

Considering the cephalometric measurements of the dental component, overbite malocclusion traits observed in 23 patients were mitigated, demonstrating the benefits of this dental interceptive orthodontic treatment in the vertical direction. On the other hand, there were no statistically significant differences between the initial and final overjet data, meaning that although the skeletal component in the sagittal direction improved, the treatment was not able to mitigate the associated overjet problem. 

As described in the literature and also verified in this sample, the expansion of the arch allows for crowding to be solved and also improves posterior dentoalveolar crossbite [22,23]. Dentoalveolar expansion with Invisalign^®^ First is designed to achieve up to 8 mm expansion. However, values between 4 and 6 mm are the most requested by clinicians [17,22]. The range of true expansions obtained in the present research has an average of 3.6 mm, within the referred literature values (Table 4). In the present research, the Invisalign^®^ First system was more successful when treating nonskeletal constricted arches, with dentoalveolar expansion correction reaching 80% with the initial series of aligners (Figure 1). In the other cases with skeletal problems and/or crossbite on the first permanent molar as an end tooth position, auxiliaries were necessary from the beginning of the treatment to achieve the crossbite correction. 

While correcting anteroposterior discrepancies such as MSC malocclusion with Invisalign^®^ Teen/Adult, some authors reported low success rates [24], consistent with the present study. Sagittal malocclusion traits with skeletal component due to mandibular retrusion are the hardest to correct, and the additional aligners requested near the official 18-month end were essential for finishing the active phase. The literature reveals that the concomitant use of auxiliaries (e.g., intermaxillary elastics) is effective in reducing treatment time in growing patients when Class II malocclusion is associated not only with dental malocclusion traits but also with mandibular retrusion [17,24,25,26,27]. The correction becomes easier if there is mandibular growing potential, as in hypodivergent biotype [28].

Considering the differences between the real and planned values of the maxillary first molar expansion width in the initial and final ClinCheck^®^, the planned corrections at the beginning of the treatment are significantly higher than those at the end of the treatment (Table 4). Of the 20 cases requiring intervention in the upper arch, 19 of them showed a decrease in the correction required, meaning that the expansion was achieved (Appendix A). Rapid maxillary expansion for opening the medial palatal suture is indicated in cases with marked transversal discrepancies between the bone bases, with a demarcated skeletal component [24]. In this sample, Invisalign^®^ First was applied only in cases where a skeletal component was associated with other difficult dental malocclusion traits. 

Another common dental malocclusion trait is the mesial rotation of the first molars, which is also analyzed here. Considering the early mixed dentition, with the second molar unerupted, the first molar derotation was considered a more complex (i.e., less accurate) movement when a correction greater than 40° was necessary. Molar derotations were also very effective and significant, with very residual or negligible corrections needed at the end of the treatment, which was expected considering that the corrections were at most on the order of 15° to 30°. 

Moreover, associated with the mesial rotation of the first molars, there usually is space loss and inherent crowding. Invisalign^®^ has proven to be an effective system for recovering space in mixed dentition, opening space for the unerupted teeth. According to the existing literature on the effectiveness of molar derotation, distalization is also one of the most accurate movements with aligners, even if the second molars have erupted [27]. In our sample, the unerupted second molars contributed to making both movements, the derotation and the distalization of the first molar, easier. 

Lastly, space recovery was attained in all 34 teeth. Space measurements performed at the end of the study are in line with the desired reference (i.e., initially planned) values. 

Overall, considering the early mixed dentition, Invisalign^®^ First was applied in complex (i.e., moderate and severe) cases. We believe that the orthodontic intervention should start with an interceptive treatment (for instance, to promote space recovery, molar derotation, and dentoalveolar expansion, among others). Of course, the orthodontic correction is inherent to the use of the aligner system as well while we are recovering the dental positioning. On the other hand, in cases with a skeletal component where no serious malocclusion traits exist, the orthopedy treatment must be chosen first [29,30].

We cannot forget that these are children with growth potential. If this growth is favorable, everything is easier. Otherwise, in cases of unfavorable growth, all biological possibilities should be considered in a way to correct the functional malocclusion traits that undoubtedly lead to an aggravation of the malocclusion trait and skeletal problems, always having in mind that these are interceptive treatments.

Although the findings was not statistically significant, the satisfaction of children relative to the use of aligners is lower than the perception of their guardians. A possible justification may be the children’s experience and adaptation to the aligners as they develop new habits and routines. However, these aligners are more comfortable, aesthetic, and hygienic compared with traditional fixed appliances [19,26,31]. This is one of the reasons for the high treatment compliance. To reinforce it, we highlight that these removable devices are particularly dependent upon the child’s collaboration to achieve treatment success. We found no evidence for a sustained rejection of the orientations provided by the orthodontist and supported by the guardians.

This article intended to analyze if the system effectiveness is related to the movement type and degree of complexity. Since this study is of a relatively unexplored orthodontic area and also considering that the Invisalign^®^ First modality started just in 2018 and comprises an 18-months interceptive treatment, our sample consisted of 23 children. However, we believe that the number of interceptive malocclusion traits (*N* = 102) remains the most representative sample for justifying the robustness of our analysis. 

## 5. Study Limitations

The convenience sample we used encompasses diverse clinical characteristics. Although the interest in the Invisalign First^®^ system has been exponentially growing, this is recent technology (available since 2018), and the number of patients with mixed dentitions using this particular system is limited. Future studies with larger sample sizes and with a randomized design are recommended. Further research using the Invisalign^®^ First system to produce tooth expansion is needed to better understand the behavior of different groups of teeth when submitted to this type of movement, mainly in children with mixed dentition. 

As we believe that the differential growth potential of the children must be considered when predicting treatment evolution, the methodology employed here cannot be compared with other studies [11,12], such as ClinCheck^®^ models’ superimposition. A controlled-randomized trial using a classification method accounting for children’s growth should be conducted to evaluate movement predictability in children having mixed dentition undergoing orthodontic treatment with Invisalign First^®^. 

The questionnaire used in the present study has not been tested for validity and reliability and could be subject to various forms of bias such as mood bias and false reporting bias among others. It is not unusual for the children to self-report more aligner wear than what occurs in reality (and unfortunately sometimes the same applies to the guardians).

## 6. Conclusions

The main points to highlight from this case series are:The interceptive treatment with Invisalign^®^ First system is effective in growing patients, being capable of producing clinical results comparable with what is planned in the ClinCheck^®^ within 18 months, although not always with the first set of aligners.The degree of complexity affects the success in mitigating various malocclusion traits, with more difficult corrections having lower efficiency rates after the first set of aligners.The statistical analysis of the measurements obtained by the ClinCheck^®^ treatment plan, table tools, millimetric scale, arch width, and cephalometric metrics at the start and end of the intervention clearly demonstrates the high effectiveness of Invisalign^®^ treatment in all domains. However, the interpretation of the ClinCheck^®^ predictions is not straightforward. Our methodology of grouping three levels of severity and assessing the percent of corrected cases allows for the extraction of meaningful clinical interpretationsHigh treatment compliance was observed in this study according to children’s and guardians’ feedback.In summary, randomized controlled trials are needed for evaluating movement predictability and how this relates to the problem complexity and the number of additional aligners required in children in early mixed dentition treated with Invisalign^®^ First.Future investigations of growing patients in early mixed dentition should use the classification methodology proposed here that considers the children’s potential growth over time.

## Figures and Tables

**Figure 1 children-09-01176-f001:**
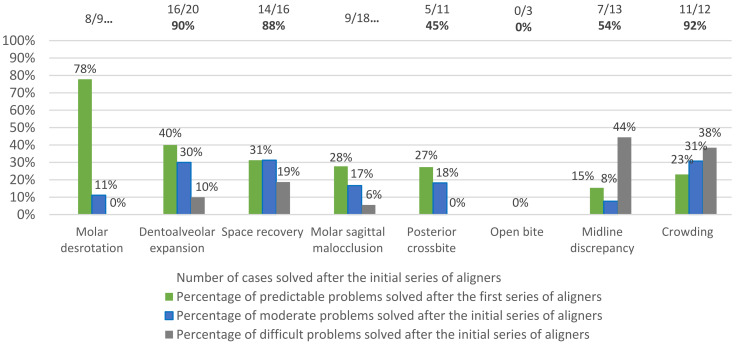
Percentages of malocclusion traits corrected after the first series of aligners according to the problem classification.

**Figure 2 children-09-01176-f002:**
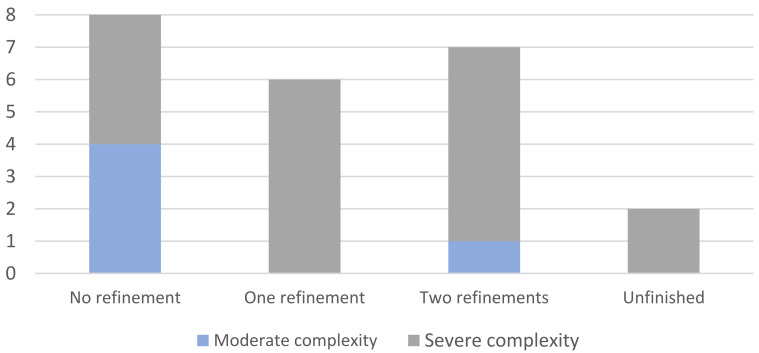
The relationships between the global complexity classification of the included cases and the number of additional aligners required to achieve all treatment objectives for the patient (*N* = 23).

**Figure 3 children-09-01176-f003:**
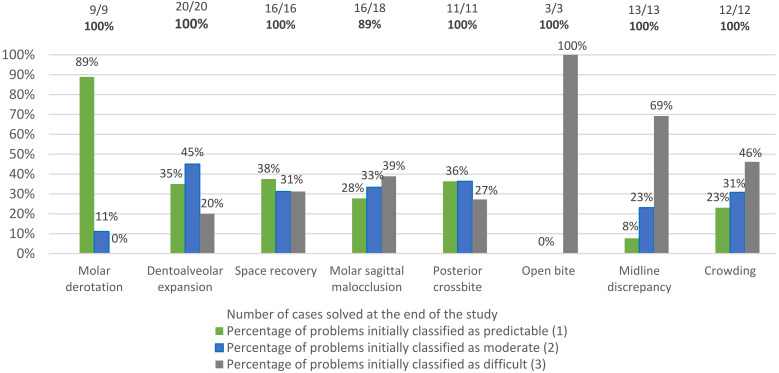
Percentages of malocclusion traits corrected at the end of the study according to the problem classification.

**Table 1 children-09-01176-t001:** Reference values for the classification of the malocclusion traits into predictable, intermediate, or difficult corrections based on Align^®^ recommendations, with the respective alterations described below.

**Malocclusion Traits** **/Required Movements**	**Type of** **Measurement**	**Predictability with Aligners**		
		Predictable (1)	Intermediate (2)	Difficult (3)
Molar derotation (teeth 16 & 26)	Initially planned by ClinCheck^®^	15–30°	>30–40°	>40°
Dentoalveolar Expansion (teeth 16–26)	Difference between the initially ClinCheck^®^ planned and experimentally measured & visual interpretation of the initial intraoral photographs	3–4 mm and Negative molar torque	>4–6 mm and Negative molar torque	>6 mm or >4 mm and Positive molar torque and Skeletal compression
Space to recover (teeth 12, 22, 32, 42, 15, 25, 33, & 43)	Difference between the initially planned by ClinCheck^®^ and experimentally measured	2–4 mm	>4–6 mm	>6 mm
Molar sagittal malocclusion	Visual interpretation of the initial intraoral photographs and ANB	Class II (incomplete or complete due to functional deviation or molar ectopic eruption)	Tendency for Class III	Class II (complete) or Class III
Posterior crossbite	Visual interpretation of the initial intraoral photographs	Crossbite only on deciduous teeth	Crossbite in deciduous/ permanent teeth and Negative molar torque	Crossbite in permanent teeth and Positive molar torque and Skeletal compression
Open bite	Initially planned by ClinCheck^®^ and FMA	Posterior intrusion: <0.5 mm and/or Anterior extrusion: <2.5 mm	Posterior intrusion: > 0.5–1 mm and/or Anterior extrusion: > 2.5–3.5 mm	Posterior intrusion: >1 mm and/or Anterior extrusion: >3.5 mm
Midline discrepancy	Initially experimentally measured	mm	>2 and <3 mm	≥3 mm
Crowding	Required space initially experimentally measured and required incisor rotation planned by ClinCheck^®^	3–6 m	>6–8 mm or Lateral incisors: 30–40° or Central incisors 40–50°	>8 mm or Lateral incisors: >40° or Central incisors: >50°

**Table 2 children-09-01176-t002:** Qualitative evaluations of each objective and the global complexity classification of every case.

	Case 1	Case 2	Case 3	Case 4	Case 5	Case 6	Case 7	Case 8	Case 9	Case 10	Case 11	Case 12	Case 13	Case 14	Case 15	Case 16	Case 17	Case 18	Case 19	Case 20	Case 21	Case 22	Case 23
Age & Sex	9.7 M	8.0 F	9.7 F	9.2 F	8.0 F	9.5 F	8.2 F	8.0 M	9.3 M	8.0 M	10.3 F	9.7 F	9.0 F	8.7 M	8.7 M	7.8 M	10.2 M	9.2 F	7.7 F	7.3 F	9.4 F	9.2 M	6.8 M
Molar derotation (*N* = 9)	0	0	0	1	1	1	0	0	0	0	1	2	0	0	0	0	0	1	0	1	1	1	0
Dentoalveolar Expansion (*N* = 20)	2	0	1	0	2	2	0	1	2	3	2	2	2	1	3	3	1	1	1	2	1	2	3
Space to recover (*N* = 16)	3	2	1	2	3	1	0	1	2	2	0	3	1	2	0	0	1	3	0	1	0	0	3
Molar sagittal Class (*N* = 18)	2	0	3	2	1	3	3	2	0	1	3	2	2	3	1	1	0	1	2	0	3	0	3
Posterior crossbite (*N* = 11)	3	1	0	0	0	2	1	2	2	0	0	0	0	0	2	3	0	0	3	1	0	1	0
Open bite (*N* = 3)	0	3	0	0	3	0	0	0	0	0	0	0	0	0	0	0	0	0	0	3	0	0	0
Midline discrepancy (*N* = 13)	3	0	0	0	0	3	0	0	3	0	1	3	3	0	2	3	0	3	2	1	3	2	3
Crowding (*N* = 13)	3	2	1	3	2	0	0	1	3	3	0	1	0	0	0	0	2	3	0	2	0	0	3
Skeletal problem (*N* = 15)	3	3	3	3	3	0	0	3	0	3	3	3	3	0	0	0	3	0	3	3	3	0	3
TOTAL	19	11	9	11	15	12	4	10	12	12	10	16	11	6	8	10	7	12	11	14	11	6	18

Caption: F: female; M: male; objectives to be achieved: Predictable (1); Intermediate (2), Difficult (3). Case global complexity:

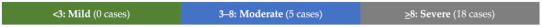

**Table 3 children-09-01176-t003:** The quantitative and qualitative assessments of the malocclusion traits to be solved for each case. These data were collected/classified at the start of the study.

	Case 1	Case 2	Case 3	Case 4	Case 5	Case 6	Case 7	Case 8	Case 9	Case 10	Case 11	Case 12	Case 13	Case 14	Case 15	Case 16	Case 17	Cas0e 18	Case 19	Case 20	Case 21	Case 22	Case 23
Molar derotation (º) (*N* = 9)				t_16 =_ 17.8 t_26 =_ 10.7	t_16 =_ 16.6 t_26 =_ 17.8	t_16 =_ 22.2 t_26 =_ 22.2					t_16 =_ 24.0 t_26 =_ 20.9	t_16 =_ 35.9 t_26 =_ 10.2						t_16 =_ 10.4 t_26 =_ 17.7		t_16 =_ 23.0 t_26 =_ 13.4	t_16 =_ 25.5 t_26 =_ 14.8	t_16 =_ 10.9 t_26 =_ 16.6	
Dentoalveolar expansion (mm) (*N* = 20)	T^-^ - 6.0		3.8		4.2	4.1		3.6	4.1	T+ 4.4	4.9	5.1	5.1	2.7	T+ 4.4	T+ 4.0	3.0	3.0	3.1	4.1	3.2	4.4	T+ 4.1
Space to recover, (mm) *N* = 16	t_12 =_ 5.5	t_12 =_ 3 t_22 =_ 2.5	t_12 =_ 2 t_22 =_ 2	t_12 =_ 2.5 t_22 =_ 2.5	t_12 =_ 3.5 t_22 =_ 4.5	t_33_ = 2.5		T_22 =_ 2.5	t_33 =_ 4.5	t_15 =_ 5.5 t_12 =_ 4 t_22 =_ 1.5 t_33 =_ 5 t_32 =_ 2 t_43 =_ 3.5		t_15 =_ 8	t_22 =_ 2.5	t_12 =_ 2.5 t_22 =_ 1			T_12 =_ 2	t_43 =_ 6,5 t_12 =_ 2.5 t_22 =_ 2.5		T_12 =_ 1.5 t_22 =_ 2			t_12 =_ 6.5 t_22 =_ 3.5 t_33 =_ 7.5 t_43 =_ 1
Molar sagittal Class, *N* = 18	Cl III tendency		Cl II complete	Cl III tendency	Cl II incomplete	Cl II complete	Cl II complete	Cl III tendency		Cl II complete; Space recovery	Cl II complete	Cl III tendency	Cl III tendency	Cl II complete	Cl II complete	Cl II incomplete		Cl II tendency	Cl III tendency		Cl II complete		Cl II complete
Posterior crossbite, *N* = 11	Skeletal t_54_, t_55_, t_16_	t_64_, t_65_				t_54_, t_55_, t_16_	t_54_, t_55_	t_26_	t_26_						T- t_64_, t_65,_ t_25_	T+ t_54_, t_55_, t_15_			Skeletal t_64_, t_65,_ t_25_	t_55/65_		t_54_	
Open bite, mm, *N* = 3		t_22 =_ 4.4E t_36 =_ 1.2I			t_21 =_ 2.6E t_16/26 =_ 1.4I															t_32 =_ 5.3E t_26 =_ 1.2I			
Midline discrepancy, mm, *N*=13	3.3					3			3.5		1.5	3	3		2.5	4		3	2		3	2	3
Crowding, mm, *N*=13	U = 8	L = 6.1	U = 4	U = 5 t_22 =_ 50.3º	L = 5.9			U = 5	L = 9	U = 9		U = 5.9					U = 5.9 t_11_ = 49.4º	L = 9		L = 5.9			L = 9
Skeletal problem *N* = 15	T	S & V ANB: 6.6 FMA: 40	S ANB: 7.8	S ANB: −1.5	T & V FMA: 32.4			S ANB: 0.9		T	S ANB: 6.4	S & T ANB: −1.6	T				S ANB: −0.1		T	V FMA: 31.9	S ANB: 5.1		S & T ANB: 6.2

Caption: Objectives to be achieved Predictable (1); Intermediate (2), Difficult (3); molar derotation values are the ones predicted at the first ClinCheck^®^; dentoalveolar expansion and space to recover values are the difference between the available space (measured) and planned at the first ClinCheck^®^; molar sagittal class malocclusions were classified according to their class; crossbite notes are about the crossed teeth; open bite values are about the intrusion or extrusion movements predicted at the first ClinCheck^®^; midline discrepancy values are the difference between the initial position of the lower and upper midlines; crowding values are the difference between the initially measured and the predicted space; skeletal problem values are the initial cephalometric parameters for the skeletal component; t_#_—tooth number; I—intrusion; E—extrusion; U—upper; L—lower; M—molar; T^+^—positive torque; T^−^—negative torque; S—sagittal; T—transversal; V—vertical; ANB—skeletal convexity; FMA—mandibular plane angle.

**Table 4 children-09-01176-t004:** Descriptive statistics regarding derotation, dentoalveolar expansion, and space recovery measurements for the teeth presenting more prominent movements to achieve, as well as cephalometry metrics for all patients.

	Mean	SD	Median	Min	Max	N
**Dental component**						9
Molar derotation (°)						
Initially planned, 16	20.7	7.9	22.2	10.4	35.9	
Planned at the end, 16	2.6	2.4	2.5	0	6.1	
Dif. (initial-final), 16	18.1	8.2	16.7	10.4	35.2	
Initially planned, 26	16.0	4.2	16.6	10.2	22.2	
Planned at the end, 26	1.7	1.8	1.1	0	4.8	
Dif. (initial-final), 26	14.4	4.3	15.4	7.6	21.1	
Dentoalveolar expansion (mm)						20
Initially measured, 16–26	43.6	2.8	43.4	38.9	48.8	
Initially planned 16–26	47.6	3.0	47.4	43.0	53.5	
Measured at the end 16–26	47.2	3.5	47.3	40.6	53.5	
Space recovery (mm)						11
Initially measured (available space), 12	3.9	1.4	4.0	1.0	6.0	
Initially planned, 12	7.1	0.6	7.0	6.0	8.0	
Measured at the end, 12	6.9	0.6	7.0	6.0	8.0	
Cephalometry (mm)						23
Overbite, initially measured	1.9	3.4	2.1	−7.0	8.6	
Overbite, measured at the end	3.3	1.1	3.2	1.3	5.7	
Overjet, initially measured	3.9	2.9	3.5	−1.6	9.4	
Overjet, measured at the end	3.6	0.9	3.4	2.4	5.4	
**Skeletal component**						
Cephalometry (°)						
ANB, initially measured	3.7	2.8	3.9	−1.6	7.9	23
ANB, measured at the end	2.7	1.9	3.2	−2.8	5.2	
FMA, initially measured	28.0	5.2	27.9	19.8	40.1	
FMA, measured at the end	27.9	4.7	27.8	16.4	36.5	

Caption: SD: standard deviation; Min: minimum; Max: maximum; N: sample size. Planned values correspond to ClinCheck^®^-predicted movements, while measured parameters are the real empirical measurements performed at each time point. To put these measurements in context, the analysis was made in subsets of a sample of 23 children with an average age of 8.3 ± 1.0 years old, 13 females and 10 males.

**Table 5 children-09-01176-t005:** The need for the improvement of the cephalometry parameters (from ANB to overjet) in relation to the reference values was compared at the beginning and at the end of the treatment. The need for correction on derotation, expansion, and space recovery metrics, comparing planned with actual measured values, was also assessed at the beginning and end of the treatment. The Wilcoxon signed-rank test was used to evaluate the progressions.

		N	Mean Rank	Z	*p*
ANB final (measured) vs. ANB initial (measured)	Negative Ranks	17 †	11.6	−2.370	0.018
	Positive Ranks	5	11.2		
	Ties	1			
	Total	23			
FMA final (measured) vs. FMA initial (measured)	Negative Ranks	8 ^†^	13.5	−0.821	0.412
	Positive Ranks	14	10.4		
	Ties	1			
	Total	23			
Overbite final (measured) vs. Overbite initial (measured)	Negative Ranks	6 ^†^	8.8	−2.586	0.010
	Positive Ranks	17	13.1		
	Ties	0			
	Total	23			
Overjet final (measured) vs. Overjet initial (measured)	Negative Ranks	12 ^†^	12.4	−0.319	0.749
	Positive Ranks	11	11.6		
	Ties	0			
	Total	23			
Derotation final (planned) vs. initial (planned), 16	Negative Ranks	8 ^†^	4.5	36.00	<0.001
	Positive Ranks	0	0.0		
	Ties	0			
	Total	8			
Derotation final (planned) vs. initial (planned), 26	Negative Ranks	7 ^†^	4.0	28.00	<0.001
	Positive Ranks	0	0.0		
	Ties	0			
	Total	7			
Expansion final (needed correction) vs. initial (needed correction), 16/26	Negative Ranks	19 ^†^	11.0	−3.85	<0.001
	Positive Ranks	1	1.0		
	Ties	0			
	Total	20			
Space recovery final (measured) vs. initial (measured)	Negative Ranks	0 ^†^	0.0	−5.12	<0.001
	Positive Ranks	34	17.5		
	Ties	0			
	Total	34			

**Caption:** ^†^ Negative ranks represent situations where the final variable measurement is lower than the initial variable measurement; positive ranks represent situations where the final variable measurement is greater than the initial variable measurement; and ties represent situations where the initial values are equal to final values.

**Table 6 children-09-01176-t006:** The relationships between the initially established degree of complexity (i.e., predictable, intermediate, or difficult corrections) and the percentage of the interceptive malocclusion traits being solved with the first series of aligners. One-sided Cochran–Armitage trend test.

	Predictable Movements		Intermediate Movements		Difficult Movements		Z	*p*
	N_solved_	N_total_	N_solved_	N_total_	N_solved_	N_total_		
Molar derotation (teeth 16 & 26)	7	8	1	1	0	0	0.375	1
Dentoalveolar Expansion (teeth 16–26)	8	8	6	9	2	3	−1.601	0.116
Space to recover (teeth 12, 22, 32, 42, 15, 25, 33 & 43)	5	6	5	5	3	5	−0.920	0.309
Molar sagittal malocclusion	5	5	3	6	1	7	−2.91	0.004
Open bite	0	0	0	0	0	3	-	-
Midline discrepancy	2	2	1	2	4	9	−1.32	0.193
Crowding	3	3	4	4	5	6	−1.002	0.459

**Table 7 children-09-01176-t007:** Responses to the questionnaires applied to parents and children in treatment with Invisalign^®^ First.

	Answer: Yes	
Group	Children (*N* = 23)	Guardian (*N* = 23)
**Question**		
1. Does the child use Invisalign for as long as the orthodontist has determined?	23 (100%)	23 (100%)
2. Does the child show interest in collaborating with the use of alignment devices?	23 (100%)	22 (96%)
3. Does the child like to use aligners?	16 (70%)	20 (87%)
4. When do you take it off?		
To sleep	1 (4%)	0 (0%)
Eat	21 (91%)	22(96%)
Play	0 (0%)	0 (0%)
School	2 (9%)	0 (0%)
Invalid	1 (4%)	1 (4%)

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
