# Peer review of "Interceptive Treatment with Invisalign® First in Moderate and Severe Cases: A Case Series"

_children, 2022, doi:10.3390/children9081176_

Round 1

Reviewer 1 Report

Dear Authors,

In this study, the authors aimed at comparing and retrospectively evaluating the predictability and effectiveness of the Invisalign® First system in orthodontic treatments in children with early mixed dentition.

The study is of scientific interest and in line with the aims of the journal, but the final number of included subjects was too small. The abstract section and the references did not respect the author guidelines. The manuscript should be copyedited by a native English speaker or copyediting service.  Moreover, the Material and Methods and Result section were not well described.

In my opinion, the manuscript is not suitable for publication in this Journal.

Main text

In the text, reference numbers should be placed in square brackets [ ], and placed before the punctuation; for example [1], [1–3] or [1,3]. For embedded citations in the text with pagination, use both parentheses and brackets to indicate the reference number and page numbers; for example [5] (p. 10). or [6] (pp. 101–105) (https://www.mdpi.com/journal/children/instructions).

Abstract

-       The abstract should be a total of about 200 words maximum. The abstract should be a single paragraph and should follow the style of structured abstracts, but without headings: 1) Background: Place the question addressed in a broad context and highlight the purpose of the study; 2) Methods: Describe briefly the main methods or treatments applied. Include any relevant preregistration numbers, and species and strains of any animals used. 3) Results: Summarize the article's main findings; and 4) Conclusion: Indicate the main conclusions or interpretations. The abstract should be an objective representation of the article: it must not contain results which are not presented and substantiated in the main text and should not exaggerate the main conclusions (https://www.mdpi.com/journal/children/instructions).

Introduction 

The Introduction section is poor. It was considered only the type of orthodontic appliance, without describing malocclusions. I suggest to improve this section, reporting the common objectives and goals of interceptive treatment, as expansion (reporting how it is recommendable – to improve crowding, preventing canine impaction, …), derotation of molars,  improving overjet and overbite, posterior crossbite resolution, etcetera..

Materials and Methods

- Where the subjects were recruited?

- When the patients were recruited?

 - The final number of patients included in the study must be reported in the Result Section, not in the Material and Method Section.

- “This study comprised a convenience sample of 23 Caucasian children (13 female and 67 10 male) having 102 interceptive problems”. Please put these information in the Result section. You have to explain where patients were recruited. Moreover, you have to report where patients were recruited.

- Line 68: “who were selected according to the following 68 inclusion criteria were considered”. This sentence is not clear, please reformulate.

- It was not reported if patients with craniofacial malformations (including cleft lip or palate), history of dental trauma, oral neoformations and other oral cavity pathologies, or previous or concomitant orthodontic treatment were excluded from the study. Please clarify the inclusion and exclusion criteria. 

- OPG and LL were used for the diagnosis? Intra and extraoral photographs? 

- Were signs or symptoms of TMJ evaluated?

- Was the study protocol approved by the Institutional Ethics Committee? Number? 

- Table 1. Predictability with aligners for the different tooth movements analyzed. Please add references about these cut-offs that you used to classy predicatibility.

- How many degrees of rotation do you perform by each aligner? How many millimiters? 

- How do you perform expansion? Did you move permanent molars buccally first, using the rest of the arch as anchorage? Or did you move deciduous and permanent molars at the same time?

- You have to put tables 2-3-4-5 in the results section (not in materials and methods!)

References did not respect the Instruction for Authors.

Author Response

Dear reviewer,

We would like to thank you for the constructive comments and suggestions. The revised manuscript includes the requested modifications and changes suggested by the reviewers (highlighted in yellow).

Dear Authors,

In this study, the authors aimed at comparing and retrospectively evaluating the predictability and effectiveness of the Invisalign® First system in orthodontic treatments in children with early mixed dentition.

The study is of scientific interest and in line with the aims of the journal, but the final number of included subjects was too small. The abstract section and the references did not respect the author guidelines. The manuscript should be copyedited by a native English speaker or copyediting service.  Moreover, the Material and Methods and Result section were not well described.

In my opinion, the manuscript is not suitable for publication in this Journal.

Authors: We acknowledge the reviewer’s time and the attention given to our manuscript. We hope the revised version of the manuscript can meet your expectations, and that you may consider it suitable for publication.

Main text

In the text, reference numbers should be placed in square brackets [ ], and placed before the punctuation; for example [1], [1–3] or [1,3]. For embedded citations in the text with pagination, use both parentheses and brackets to indicate the reference number and page numbers; for example [5] (p. 10). or [6] (pp. 101–105) (https://www.mdpi.com/journal/children/instructions).

Authors: Thank you for this suggestion. The requested modification was applied in the revised version of the manuscript.

 Abstract

-       The abstract should be a total of about 200 words maximum. The abstract should be a single paragraph and should follow the style of structured abstracts, but without headings: 1) Background: Place the question addressed in a broad context and highlight the purpose of the study; 2) Methods: Describe briefly the main methods or treatments applied. Include any relevant preregistration numbers, and species and strains of any animals used. 3) Results: Summarize the article's main findings; and 4) Conclusion: Indicate the main conclusions or interpretations. The abstract should be an objective representation of the article: it must not contain results which are not presented and substantiated in the main text and should not exaggerate the main conclusions (https://www.mdpi.com/journal/children/instructions).

Authors: We thank the reviewer for the above comment, and we have now modified the Abstract according to the reviewer’s instructions.  

Introduction 

The Introduction section is poor. It was considered only the type of orthodontic appliance, without describing malocclusions. I suggest to improve this section, reporting the common objectives and goals of interceptive treatment, as expansion (reporting how it is recommendable – to improve crowding, preventing canine impaction, …), derotation of molars,  improving overjet and overbite, posterior crossbite resolution, etcetera..

Authors: We appreciate the reviewer’s attention to this point. The Introduction was altered accordingly (page 1-2, lines 38-41 & lines 43-51).

 Materials and Methods

- Where the subjects were recruited?

Authors: The patients were recruited in two independent clinics of orthodontics – Clínica Médico Dentária de São João da Madeira e Clínica de Medicina Dentária Dr. Manuel Neves. This information were added to the first paragraph of this section (page 2, lines 76-80).

- When the patients were recruited?

Authors: The patients were recruited from October 2018 to October 2019, and this information were added to the manuscript (page 2, lines 78).

 - The final number of patients included in the study must be reported in the Result Section, not in the Material and Method Section.

Authors: Thank you for this suggestion. We have considered this commentary and made the respective modification.

- “This study comprised a convenience sample of 23 Caucasian children (13 female and 67 10 male) having 102 interceptive problems”. Please put these information in the Result section. You have to explain where patients were recruited. Moreover, you have to report where patients were recruited.

Authors: Thank you. Modifications were done accordingly.

- Line 68: “who were selected according to the following 68 inclusion criteria were considered”. This sentence is not clear, please reformulate.

Authors: We appreciate the reviewer’s attention to this detail. We have reformulated this sentence (page 2, lines 81-86).

- It was not reported if patients with craniofacial malformations (including cleft lip or palate), history of dental trauma, oral neoformations and other oral cavity pathologies, or previous or concomitant orthodontic treatment were excluded from the study. Please clarify the inclusion and exclusion criteria. 

Authors: Thank you for highlighting this point. All the mentions cases were excluded, and this information is now provided in the revised version of the manuscript (page 2, lines 86-89).

- OPG and LL were used for the diagnosis? Intra and extraoral photographs? 

Authors: The referred biomarkers were not evaluated for diagnosis in this study. On the contrary, virtual three-dimensional (3D) planning using the ClinCheck® software, iTero® intraoral digital models and intraoral photographs were also obtained from each patient. This information is now more clearly depicted in the revised version of the manuscript (page 5, lines 184-190).

- Were signs or symptoms of TMJ evaluated?

Authors: Yes, signs and symptoms of temporomandibular disorders were clinically assessed, and no dysfunctions were seen. This information were added to the manuscript (page 5, lines 195-196; page 6, line 223-224).

- Was the study protocol approved by the Institutional Ethics Committee? Number? 

Authors: Yes, and we have provided the Ethics Committee approval document to the journal. The respective reference (11/CE-IUCS/2020) is now indicated in the manuscript (page 2, line 93).

- Table 1. Predictability with aligners for the different tooth movements analyzed. Please add references about these cut-offs that you used to classy predicatibility.

Authors: As referred in the manuscript, the movement classification was performed based on an adaptation of the Align® protocol, considering the potential growth of the children over time. A reference to the protocol is provided in the manuscript. In addition, we thoroughly describe the criteria used to evaluate each type of problem (pages 4-5, lines 121-174).

- How many degrees of rotation do you perform by each aligner? How many millimiters? 

Authors: It is not possible to quantify the amount of movement predicted for in each aligner. Each aligner is designed to produce a 0.25-mm translation and 1° of rotation movement for standard forces. This information were added to the revised version of the manuscript (page 2, lines 96-97).

- How do you perform expansion? Did you move permanent molars buccally first, using the rest of the arch as anchorage? Or did you move deciduous and permanent molars at the same time?

Authors: Usually, we primarily expand the permanent molars and then the deciduous teeth to serve as anchorage.

- You have to put tables 2-3-4-5 in the results section (not in materials and methods!)

Authors: We would like to thank the reviewer for his/her thorough examination and assertive suggestions. All these issues are now corrected or better described in the new version of the manuscript.

References did not respect the Instruction for Authors.

Authors: We would like to thank the reviewer’s attention to this point. We have corrected this issue.

Reviewer 2 Report

This study holds high clinical value for orthodontists who use the Invisalign First systems and the research is detailed and thorough.

In the introduction it would be helpful if you described the ClinCheck system as well for a better understanding of the study

Minor English language and phrasing check is required

The results could be structured to be a bit clearer and less convoluted 

Author Response

Dear reviewer,

We would like to thank you for the constructive comments and suggestions. The revised manuscript includes the requested modifications and changes suggested by the reviewers (highlighted in yellow).

This study holds high clinical value for orthodontists who use the Invisalign First systems and the research is detailed and thorough.

In the introduction it would be helpful if you described the ClinCheck system as well for a better understanding of the study.

Authors: We would like to thank the reviewer for his/her constructive comments and the appreciation of the novel aspect for our review. More detailed information about the ClinCheck® system was now added to the revised version of the manuscript (pages 1 and 2, lines 54-55).

Minor English language and phrasing check is required.

Authors: Thank you for this suggestion. Linguistic and grammar issues have been carefully checked and corrected in the revised manuscript.  

The results could be structured to be a bit clearer and less convoluted.

Authors: We acknowledge the reviewers’ comments and we took it into consideration in the new version of the manuscript. Indeed, the results were previously too dense and hard to read. We now added subheading to this section and more focused and assertive information is provided. We hope this meets the reviewer’s expectations.

Reviewer 3 Report

Dear authors, 

I appreciate the effort to make this type of clinical research, which I consider of extreme importance in our area, is the use of aligners a hot topic. However, I have some concerns and suggestions about the study:

Introduction

- Minor revisions to English use. Avoid long sentences. Review word choice and preposition use.

- The study's aims are unclear; use the same definitions along with the text. What do you want to say with problems/pathologies? Malocclusion traits?

 Material & Methods

- The statement "This study comprised a convenience" implies a high selection bias. 

Please consider presenting this in another way. Also, include how many patients did not comply with the Invisalign treatment. 

- You present in the line 67 and 74 results of the study. Just limit to material and methods.

-What do you mean by "The required movements included…." You have table 1. 

- 2.2 Ethical considerations: please provide the number of approval from the Ethics Committee. Please take an account the choice word. "Study onset", I think you mean this by "the start of the treatment." All this information provided from lines 87-92 is unnecessary. The Ethics Committee is supposed to review and evaluate a study protocol, where the patients' informed consent is included. So you do not need to write this here. 

- I suppose you mean the last phase as the "retention phase". 

- By 2.3 clinical assessment: please limit the content to the material & methods and avoid presenting the results in this section. I suggest to include/ combine this information in table 1.  

- Again, table 2, must be in the results section. 

- Page 7, you have to describe exactly what you did with the data. Table 3 is not to follow. 

- I suggest that you have to make a choice. Maybe you can concentrate only  in occlusion traits, or in cases as moderate and severe as state in the title of your study.

- In section 2.5, which statistical tests were performed? If you used studio R and IBM SPSS, which analysis do you have to perform in Excel? 

Please be aware that from here, you are presenting the results. This information has to be in the following section. Please be aware that you haven't mentioned anything about the questionary to the parents. Which questions were included? When it was it given to the parents?

- By table 4

  • Please complete table 4 with descriptive data of the population, sex distribution, age. 
  • Molar derotation, what do you want to say with: "Molar derotation with : Initially planned, 16, planned at the end, 16 Initially planned, 26 Planned at the end, 26…. What is the description? It is not easier to speak about the reduction of the molar rotation at the end of the treatment? 
  • About the skeletal component, why did you measure this only in 8 patients? 
  • ANB alone, does not give much information. 
  • Very difficult to follow what you did the Wilcoxon signed-rank test en what you want to proof of demonstrate with this test. 
  • If the parent's questionary hasn't been validated, consider to remove from the study, seems any conclusion can be taken. Maybe you can use the data for your use and later validation. 

Results 

- Difficult to follow. I didn't go into detail since previous sections need improvement. See the last comments. This section is challenging to follow. 

Why in table 6 are other cephalometric parameters not mentioned before? 

- figure 1, unclear.

Discussion

- overestimate result, since only patients with good compliance were followed

- what do you mean by malocclusion? It may be better to refer as malocclusion traits. 

- I didn't go into detail since previous sections need improvement.

I acknowledged the significant effort to present this study. However, I consider it has substantial problems. I suggest using the comments and rewriting the article. Maybe limit the survey to only evaluating occlusion traits before and after the treatment. Describe accurately material and methods, and present the results only in the "results section." 

Success, 

Author Response

Dear reviewer,

We would like to thank you for the constructive comments and suggestions. The revised manuscript includes the requested modifications and changes suggested by the reviewers (highlighted in yellow).

Dear authors, 

I appreciate the effort to make this type of clinical research, which I consider of extreme importance in our area, is the use of aligners a hot topic. However, I have some concerns and suggestions about the study:

Introduction

- Minor revisions to English use. Avoid long sentences. Review word choice and preposition use.

Authors: We acknowledge the reviewer for this comment. Linguistic and grammar issues have been carefully checked and corrected in the revised manuscript.

- The study's aims are unclear; use the same definitions along with the text. What do you want to say with problems/pathologies? Malocclusion traits?

Authors: Thank you for this suggestion. The aim of the study is now clearly described in the Abstract and Introduction sections (page 1, lines 21-22; page 2, lines 68-73). Moreover, we  added a sentence describing the definition of the interceptive problems reviewed in the current study (page 1, lines 38-41), and we believe that the terminology is not confusing anymore.

  Material & Methods

- The statement "This study comprised a convenience" implies a high selection bias. 

Please consider presenting this in another way. Also, include how many patients did not comply with the Invisalign treatment.

Authors: We appreciate the reviewer’s constructive comment. We have now rephrased this sentence (pages 6, lines 222-223).

All patients initially selected for this study completed the orthodontic treatment, and no drop-outs occurred (page 6, lines 228-229).

- You present in the line 67 and 74 results of the study. Just limit to material and methods.

Authors: Thank you for this comment. We have performed the required changes.

-What do you mean by "The required movements included…." You have table 1. 

Authors: Thank you for this suggestion. That information was rewritten and transferred to the Results section (page 6, lines 225-228).  

- 2.2 Ethical considerations: please provide the number of approval from the Ethics Committee.

Authors: Yes, and we have provided the Ethics Committee approval document to the journal. The respective reference (11/CE-IUCS/2020) is now indicated in the manuscript (page 2, line 93).

Please take an account the choice word. "Study onset", I think you mean this by "the start of the treatment."

Authors: Thanks for this suggestion. It was taken into consideration in the revised version of the manuscript.

All this information provided from lines 87-92 is unnecessary. The Ethics Committee is supposed to review and evaluate a study protocol, where the patients' informed consent is included. So you do not need to write this here. 

Authors: Thank you for this suggestion. That unnecessary information was removed.

- I suppose you mean the last phase as the "retention phase". 

Authors: Actually, the last phase refers to the last additional aligners considered when the objectives were not achieved yet, to perform overcorrections or other final adjustment.

- By 2.3 clinical assessment: please limit the content to the material & methods and avoid presenting the results in this section. I suggest to include/ combine this information in table 1.  

Authors: Thank you for this suggestion. We have considered this commentary and made the respective modifications.

- Again, table 2, must be in the results section. 

Authors: Thank you. This was moved to the Results section.

- Page 7, you have to describe exactly what you did with the data. Table 3 is not to follow. 

Authors: We appreciate the reviewer’s attention to this point. This table can be interpretated on its own. The caption provides all the information required to analyze the data and to interpret all values, for each problem.

- I suggest that you have to make a choice. Maybe you can concentrate only in occlusion traits, or in cases as moderate and severe as state in the title of your study.

Authors: The authors acknowledge this recommendations. This case series only focus on moderate and severe cases according to the global complexity classification depicted in Table 2. This information in now more clearly described in the revised version of the manuscript (page 6, lines 235-238).

- In section 2.5, which statistical tests were performed? If you used studio R and IBM SPSS, which analysis do you have to perform in Excel? 

Authors: The statistical tests used in the study (i.e., Fisher’s exact test, one-sided Cochran-Armitage Trend test, Bonferroni correction for multiple testing, and Wilcoxon signed-rank test) are described in section 2.5, as mentioned by the reviewer. Excel was used for graphical reporting. This information was added to the new version of the manuscript (page 5, lines 201-202).

Please be aware that from here, you are presenting the results. This information has to be in the following section. Please be aware that you haven't mentioned anything about the questionary to the parents. Which questions were included? When it was it given to the parents?

Authors: Thank you for your comments. We have considered them and we’ve made the respective modifications.

The questionnaire is depicted in Table 8, and it was applied to parents and patients.

- By table 4

  • Please complete table 4 with descriptive data of the population, sex distribution, age. 

Authors: We appreciate the reviewer’s suggestion, but this table refers to the descriptive data of the clinical outputs, and not demographics. Nevertheless, in order to explain the context of the measurements on the target population, the information on average age and gender was added to the caption of Table 4.

  • Molar derotation, what do you want to say with: "Molar derotation with: Initially planned, 16, planned at the end, 16 Initially planned, 26 Planned at the end, 26…. What is the description? It is not easier to speak about the reduction of the molar rotation at the end of the treatment? 

Authors: Thank you for your suggestion. We compared the molar rotation planned at the beginning of the study (at the first ClinCheck) with the rotation that stills being planned at the end of the study (at the last ClinCheck). Nevertheless, we added the descriptive data of the difference between these two values, as suggested, in order to easily compare the magnitude of reduction of the needed rotation for each tooth. 

  • About the skeletal component, why did you measure this only in 8 patients? 

Authors: The analysis only included the patients presenting each problem (i.e., molar derotation, dentoalveolar expansion, space recovery, or cephalometry). Only eight patients presented skeletal problems in the sagittal direction (evaluated using ANB and overjet), while only three patients had skeletal problems in the vertical direction, evaluated through FMA and overbite. This information is described in Table 3.

  • ANB alone, does not give much information. 

Authors: ANB allows the identification of an alveolar problem in the sagittal direction. Moreover, overjet also complements the diagnosis as it provides information on the dental component in the same direction. 

  • Very difficult to follow what you did the Wilcoxon signed-rank test en what you want to proof of demonstrate with this test. 

Authors: We appreciate the reviewer’s attention to this point. The information provided in Table 5 is in accordance with the recommendations for reporting the statistics of the Wilcoxon signed-rank test. We have reformulated the analysis of the referred table within the text to facilitate the interpretation of the data. 

  • If the parent's questionary hasn't been validated, consider to remove from the study, seems any conclusion can be taken. Maybe you can use the data for your use and later validation. 

Authors: We thank the reviewer for pointing out this issue. These questionnaires only intended to ascertain if the patients committed and complied to the treatment. Importantly, at the time of development of the current study, no questionnaires were validated for children in these ages, and therefore these kind of dysfunctions were not addressed through this method.

 Nevertheless, if the reviewer think this information must be removed, we can do it so. However, we think this is an important information on treatment compliance and satisfaction.

Results 

- Difficult to follow. I didn't go into detail since previous sections need improvement. See the last comments. This section is challenging to follow. 

Authors: We acknowledge the reviewers’ comments and we took it into consideration in the new version of the manuscript. Indeed, the results were previously too dense and hard to read. We now added subheading to this section and more focused and assertive information is provided. We hope this meets the reviewer’s expectations.

Why in table 6 are other cephalometric parameters not mentioned before? 

Authors: The authors appreciate the reviewer’s attention to this issue. The referred parameters were removed.

- figure 1, unclear.

Authors: Thank you for your point. Line colors were changed according to Tables 2 and 3. Moreover, this intends to describe relation between the percentage of cases solved after the initial series of aligners and the complexity of the problems. The analysis of this figure is presented in page 11, lines 329-334 & lines 338-341.

Discussion

- overestimate result, since only patients with good compliance were followed

Authors: All patients included in the study showed high compliance. As a case series, we provide indications for future studies, recalling that controlled-randomized clinical trials are required to obtain more reliable data and ascertain the predictability of Invisalign First in mixed dentition.

- what do you mean by malocclusion? It may be better to refer as malocclusion traits. 

Authors: Thank you for this suggestion. We now clearly describe the definition of the interceptive problems reviewed in the current study (page 1, lines 34-37), and we believe that the terminology is not confusing anymore. The term “malocclusion” was replaced by “malocclusion traits” when appropriate.

I didn't go into detail since previous sections need improvement.

I acknowledged the significant effort to present this study. However, I consider it has substantial problems. I suggest using the comments and rewriting the article. Maybe limit the survey to only evaluating occlusion traits before and after the treatment. Describe accurately material and methods, and present the results only in the "results section." 

Success, 

Authors: We acknowledge the reviewer’s time and the attention given to our manuscript. We hope the revised version of the manuscript can meet your expectations, and that you may consider it suitable for publication.

Round 2

Reviewer 1 Report

Dear Authors,

In this study, the authors aimed at comparing and retrospectively evaluating the predictability and effectiveness of the Invisalign® First system in orthodontic treatments in children with early mixed dentition.

The study is of scientific interest and in line with the aims of the journal, but the final number of included subjects was too small. The abstract section and the references did not respect the author guidelines. The manuscript should be copyedited by a native English speaker or copyediting service.  Moreover, the Material and Methods and Result section were not well described.

In my opinion, the manuscript is not suitable for publication in this Journal.

Author Response

Dear reviewer,

We would like to thank you for your constructive comments and suggestions. The revised manuscript includes the requested modifications and changes suggested by the reviewers. The current revisions are highlighted in green and previous revisions are in yellow.

Reviewer 1

Dear Authors,

In this study, the authors aimed at comparing and retrospectively evaluating the predictability and effectiveness of the Invisalign® First system in orthodontic treatments in children with early mixed dentition.

The study is of scientific interest and in line with the aims of the journal, but the final number of included subjects was too small.

Authors: We acknowledge the reviewer’s time and the attention given to our manuscript. Case reports have often less patients than the current paper. Besides, although we have included 23 patients, these had 102 interceptive orthodontic problems, which we believe is a considerable sample.

The abstract section and the references did not respect the author guidelines. 

Authors: We appreciate the reviewer’s attention to this point, and we have corrected it accordingly.

The manuscript should be copyedited by a native English speaker or copyediting service.  

Authors: Thank you for this feedback. Linguistic and grammar issues have been carefully checked and corrected by a native English speaker. 

Moreover, the Material and Methods and Result section were not well described.

Authors: We appreciate the reviewer’s attention to this point. The referred sections were modified, and now we believe the manuscript is more focused and easy-to-follow.

In my opinion, the manuscript is not suitable for publication in this Journal.

Authors: We hope the revised version of the manuscript can meet your expectations, and that you may consider it suitable for publication.

Reviewer 3 Report

Dear authors,

I really acknowledge the effort to correct the manuscript. It has significantly improved. However, I believe the manuscript needs major revisions, as my previous comments did. I do not think the manuscript is ready for scientific publication. Please also be aware that last time I concentrate on the first sections as I do this time.

In this version, the article is hard to read. The English needs major corrections. For instance, In the abstract on page 1, line 24, when you refer to "First were followed-up and examined for 18 months since the treatment start," it is unclear. First were the patients follow-up? for how much time? and then examinated? it doesnt make sense.

Also, as mentioned in my previous letter, the word 'onset' is inadequate. Therefore, the statement: "Planned and real metrics acquired in the onset and end were compared" is unclear. Planned and real metrics? What do you mean? You mean the planed treatment with compared with the obtained results after finish the treatment.

1. Introduction

- the statement "Their early diagnosis is essential to create 41 the optimal conditions for the execution of treatment plans and subsequently to allow more effective interceptive treatments."  is inaccurate. A correct diagnosis is essential for all types of treatments; maybe you want to say, "The early diagnosis is essential to prevent future extensive orthodontic treatment."

- "First system can be re-activated..", line 57. You have to be more specific. Do you mean a refinement?

- Still do not agree with the words: problems/pathologies. Maybe you want to say maloocclusion traits…

2. Material and methods

- Please avoid person names in the text. For example, you can write that all patients were treated by one experienced specialist (TP).

- When you stated that children only children who were committed to the treatment. You have a severe bias, and all results are overestimated. Consider presenting a chart with the population selection because clinicians need to know how many patients are not compliant with the treatment. If you state that you only choose patients with compliance, it implies that you also have patients who stopped with the treatment.

- Line 96-97: "Each aligner is designed to produce a 0.25-mm translation and 1° of rotation movement for standard forces (Align 2022).|" What do you want to state with this sentence? Which point do you want to make if this prescription is commonly used from Align?

-This last phase was fundamental to consolidate the good results achieved within the official period'' what do you refer to? The retention?

- Table 1:

Be aware that the ranges are inappropriate. For instance, the values of 30° and 40° are included as predictable as moderate.

I think you mean: predictable 15-30°, Moderate >30-40, Difficult: >40°.

Please review the table and correct it.

-Line 116-118: unclear. Which reference values?

-You start here with a classification? Which classification? What is it about? It is not clear. Also, on lines 121- 174, again, results are included in this section. For instance, You state that in molar rotations, 9 cases were identificated. So you do not have to write 'n' numbers here.

-Then, in line 176 you start "base on this qualitative evaluation", Was it a classification?. It is not clear how you come now to this statement

-Line 184, you mention "multiple methods were used to analyse the included cases"…Which methods? Here it would help if you were more specific.

In this section, you have to restrict to describe the material and methods.

After reading this section, it is still unclear who and how the data was collected, who and how was made the measurements (by super imposition of two digital models), which records were used. Finally, how many times was de data measurement repeated and which is the intraclass correlation coefficient.

3.Results

Here are some corrections. I did not go into detail.

-Line 235-238 explains the classification you mentioned in the previous section.

-Table 2, first row in Spanish.

- Figure 1 and 3 . Why do you use a connecting line between all malocclusion traits? Please, you need to present the data differently because it conducts to misinterpretations.

I know how much effort it takes to collect data and research, but I think you must work significantly to improve. Unfortunately, the material and methods section still has serious problems. I didn't continue further with the revision of other manuscript sections. I encourage you to rethink how to present all the information; what is essential for the clinicians. What do you want to research? I think you have to make a choice. Do you want to investigate how efficient clear aligners are on molar rotation, dentoalveolar expansion, crowding, etc? Or do you want to see how efficient it is to resolve moderate and severe cases? Because I think it is not clear. At some point, the reader cannot follows you anymore. So much information is presented in such a way that it makes it difficult to understand. I think you have incredible data; you only need to put it together.

Best regards,

Author Response

Dear reviewer,

We would like to thank you for your constructive comments and suggestions. The revised manuscript includes the requested modifications and changes suggested by the reviewers. The current revisions are highlighted in green and previous revisions are in yellow.

Reviewer 3

Dear authors,

I really acknowledge the effort to correct the manuscript. It has significantly improved. However, I believe the manuscript needs major revisions, as my previous comments did. I do not think the manuscript is ready for scientific publication. Please also be aware that last time I concentrate on the first sections as I do this time.

In this version, the article is hard to read. The English needs major corrections. For instance, In the abstract on page 1, line 24, when you refer to "First were followed-up and examined for 18 months since the treatment start," it is unclear. First were the patients follow-up? for how much time? and then examinated? it doesnt make sense.

Authors: We appreciate the reviewer’s comment. Linguistic and grammar issues have been carefully checked and corrected by a native English speaker. Also, we have reformulated the sentence mentioned by the reviewer (page 1, line 23-25).

Also, as mentioned in my previous letter, the word 'onset' is inadequate. Therefore, the statement: "Planned and real metrics acquired in the onset and end were compared" is unclear. Planned and real metrics? What do you mean? You mean the planed treatment with compared with the obtained results after finish the treatment.

Author: Thank you for pointing out this issue. We speak about two kind of distance/rotation metrics – ClinCheck®-planned or ClinCheck®-predicted values, which were evaluate both at the beginning and at the end of the treatment (Phase 1); and the real  amounts of movement/space measured experimentally, which was also assessed at the start and at the end of the treatment. This results into four movements – planned at the beginning, planned at the end, experimentally measured at the beginning and experimentally measured at the end. We have clarified this issue (page 6, lines 250-253), and the expressions used to refer to these movements were uniformized throughout the new version of the manuscript.   

  1. Introduction

- the statement "Their early diagnosis is essential to create 41 the optimal conditions for the execution of treatment plans and subsequently to allow more effective interceptive treatments."  is inaccurate. A correct diagnosis is essential for all types of treatments; maybe you want to say, "The early diagnosis is essential to prevent future extensive orthodontic treatment."

Author: Thank you very much for this suggestion. This is exactly what we intended to say, and it was corrected accordingly the revised manuscript.

- "First system can be re-activated..", line 57. You have to be more specific. Do you mean a refinement?

Author: We appreciate the reviewer’s attention to this point. Indeed, Phase 2 is not a refinement. The normal orthodontic interceptive treatment with the Invisalign First system, commonly called Phase 1, has an 18-months duration. After ending Phase 1, the orthodontic treatment can be re-activated with a new treatment (Phase 2) with a duration of three years maximum. This could occur within ten years since Phase 1 ends if the malocclusion trait was not resolved or in case of a recurrence. In the present study, the predictability was only assessed on Invisalign First during Phase 1 (page 2, lines 55-59).

- Still do not agree with the words: problems/pathologies. Maybe you want to say malocclusion traits…

Author: We have corrected this expression throughout the manuscript.

  1. Material and methods

- Please avoid person names in the text. For example, you can write that all patients were treated by one experienced specialist (TP).

Author: Thank you for highlighting this point. We have made de requested change in the text and deleted the person name.

- When you stated that children only children who were committed to the treatment. You have a severe bias, and all results are overestimated. Consider presenting a chart with the population selection because clinicians need to know how many patients are not compliant with the treatment. If you state that you only choose patients with compliance, it implies that you also have patients who stopped with the treatment.

Author: We really appreciate the reviewer’s attention to this issue. High compliance was not an inclusion criteria, but an output we have collected from the children and their parents. Indeed, from the patients selected in the beginning of the treatment, none were excluded for low commitment or any other motif. Therefore, high compliance was removed from the inclusion criteria (page 2, lines 83-89). 

- Line 96-97: "Each aligner is designed to produce a 0.25-mm translation and 1° of rotation movement for standard forces (Align 2022).|" What do you want to state with this sentence? Which point do you want to make if this prescription is commonly used from Align?

Author: We appreciate your opinion on this point. The referred topic was added in the last review since one of the reviewers (reviewer 1) asked “How many degrees of rotation do you perform by each aligner? How many millimiters? “. Thus, we have clarified this issued and added the respective citation. Moreover, this is like a reference value that can be used to evaluate the efficiency of the system, and so we believe this is an important information to provide.

-This last phase was fundamental to consolidate the good results achieved within the official period'' what do you refer to? The retention?

Author: Yes, retention. It was only required to solve the two Class II situations that were not fully corrected within the 18 months (page 3, lines 109-112).

Table 1: Be aware that the ranges are inappropriate. For instance, the values of 30° and 40° are included as predictable as moderate.

I think you mean: predictable 15-30°, Moderate >30-40, Difficult: >40°.

Please review the table and correct it.

Author: We appreciate the reviewer’s attention to this point. We have made the correction in the manuscript.

-Line 116-118: unclear. Which reference values?

Author: Table 1 contains the reference values for the classification of the malocclusion traits into predictable, intermediate and difficult based on an adaptation of Align® recommendations for the classification of interceptive problems.

-You start here with a classification? Which classification? What is it about? It is not clear. Also, on lines 121- 174, again, results are included in this section. For instance, You state that in molar rotations, 9 cases were identificated. So you do not have to write 'n' numbers here.

Author: Thank you for this recommendation. Those numbers were results and therefore they were removed from the referred excerpt. Moreover, the two classification methodologies we have used (one for malocclusions traits and another for cases) are clearly describe (Table 1; pages 5-6, lines 149-247).

-Then, in line 176 you start "base on this qualitative evaluation", Was it a classification?. It is not clear how you come now to this statement

Author: Thank you for this topic. We have corrected the term “evaluation” to “classification”.

  • Line 184, you mention "multiple methods were used to analyse the included cases"…Which methods? Here it would help if you were more specific.

Author: Thank you for this observation. We now clearly state that the methods utilized for the clinical assessment (page 6-7, lines 245-266).

In this section, you have to restrict to describe the material and methods.

Author: Thank you for your comment. We revised this section in the new version of the manuscript.

After reading this section, it is still unclear who and how the data was collected, who and how was made the measurements (by super imposition of two digital models), which records were used. Finally, how many times was de data measurement repeated and which is the intraclass correlation coefficient.

Authors: We appreciate the reviewer’s comment. In the current study, having the same observer classifying each subject at different times, as well as different observers classifying the cases within the same period, the same classifications were given. With that said, we didn’t feel the need to include data on intra and inter observer reliability because all the classifications were independent and they all obtained the same result. We have added this idea on the manuscript (page 3, lines 116-118) 

  1. Results

Here are some corrections. I did not go into detail.

-Line 235-238 explains the classification you mentioned in the previous section.

Author: We appreciate the reviewer’s attention to this point. We have only reinforced how the cases were classified, adding the information on the number of cases classified as moderate and severe (page 7, lines 296-299).

-Table 2, first row in Spanish.

Author: Thank you for your attention to this detail. This was corrected in the new version of the manuscript.

- Figure 1 and 3 . Why do you use a connecting line between all malocclusion traits? Please, you need to present the data differently because it conducts to misinterpretations.

Authors: Thank you for this tip. Indeed, data labels were overlapping each other and the lines. Moreover, we have changed the graph type since the connecting lines were misleading. We believe the new Figures 1 and 3 are quite easier to analyze.

 I know how much effort it takes to collect data and research, but I think you must work significantly to improve. Unfortunately, the material and methods section still has serious problems. I didn't continue further with the revision of other manuscript sections. I encourage you to rethink how to present all the information; what is essential for the clinicians. What do you want to research? I think you have to make a choice. Do you want to investigate how efficient clear aligners are on molar rotation, dentoalveolar expansion, crowding, etc? Or do you want to see how efficient it is to resolve moderate and severe cases? Because I think it is not clear. At some point, the reader cannot follows you anymore. So much information is presented in such a way that it makes it difficult to understand. I think you have incredible data; you only need to put it together.

 Best regards.

Author: These various problems to solve are dependent of each other, not being correct to consider them separately as other investigations do. In this paper we intended to correlate them.

We hope the revised version of the manuscript can meet your expectations, and that you may consider it suitable for publication.
